# Computationally inferred cell-type specific epigenome-wide DNA methylation analysis unveils distinct methylation patterns among immune cells for HIV infection in three cohorts

Xinyu Zhang[1,2], Ying Hu[3], Ral E. Vandenhoudt[1,2], Chunhua Yan[3], Vincent C. Marconi[4,5], Mardge H. Cohen[6], Zuoheng Wang[7], Amy C. Justice[2,7,8], Bradley E. Aouizerat[9,10☉]*, Ke Xu [1,2,11☉]*

1 Department of Psychiatry, Yale School of Medicine, New Haven, Connecticut, United States of America, 2 VA Connecticut Healthcare System, West Haven, Connecticut, United States of America, 3 Center for Biomedical Information and Information Technology, National Cancer Institute, Rockville, Maryland, United States of America, 4 Division of Infectious Diseases, Emory University School of Medicine and Department of Global Health, Rollins School of Public Health, Emory University, Georgia, United States of America, 5 Atlanta Veterans Affairs Healthcare System, Decatur, Georgia, United States of America, 6 Department of Medicine, Stroger Hospital of Cook County, Chicago, Illinois, United States of America, 7 Department of Biostatistics, Yale School of Public Health, New Haven, Connecticut, United States of America, 8 Department of Internal Medicine, Yale School of Medicine, New Haven, Connecticut, United States of America, 9 Translational Research Center, College of Dentistry, New York University, New York, New York, United States of America, 10 Department of Oral and Maxillofacial Surgery, College of Dentistry, New York University, New York, New York, United States of America, 11 Biomedical Informatics and Data Science, Yale School of Medicine, New Haven, Connecticut, United States of America

☉ These authors contributed equally to this work.
* bea4@nyu.edu (BEA); ke.xu@yale.edu (KX)

**Data Availability Statement:** Demographic and clinical variables and DNAm data for the VACS

## Abstract

### Background

Epigenome-wide association studies (EWAS) have identified CpG sites associated with HIV infection in blood cells in bulk, which offer limited knowledge of cell-type specific methylation patterns associated with HIV infection. In this study, we aim to identify differentially methylated CpG sites for HIV infection in immune cell types: CD4+ T-cells, CD8+ T-cells, B cells, Natural Killer (NK) cells, and monocytes.

### Methods

Applying a computational deconvolution method, we performed a cell-type based EWAS for HIV infection in three independent cohorts ($N_{total}$ = 1,382). DNA methylation in blood or in peripheral blood mononuclear cells (PBMCs) was profiled by an array-based method and then deconvoluted by Tensor Composition Analysis (TCA). The TCA-computed CpG methylation in each cell type was first benchmarked by bisulfite DNA methylation capture sequencing in a subset of the samples. Cell-type EWAS of HIV infection was performed in each cohort separately and a meta-EWAS was conducted followed by gene set enrichment analysis.

samples were submitted to GEO dataset (GSE117861) and are publicly available. All codes for analysis are also available at https://github.com/KeLab2018/Deconvoluted_HIV_EWAS.

**Funding:** The project was supported by the National Institute on Drug Abuse (NIDA; R03DA039745, R01DA047063, R01DA047820, R01DA038632 to KX, and U01 AA020790 to ACJ); by the National Institute on Alcohol Abuse and Alcoholism (NIAA, a VACS award from which the majority of our cohort data is sourced; U10 AA013566-completed to KX), and by the COMpAAAS coordinating center funded by the NIAA (U24 AA020794 to ACJ). COMpAAAS/ Veterans Aging Cohort Study, a CHAART Cooperative Agreement, is supported by the NIAAA (U24-AA020794, U01-AA020795, U01-AA020799; U10-AA013566-completed to ACJ) and in kind by the US Department of Veterans Affairs. Additional grant support from the National Institute on Drug Abuse R01-DA035616 is acknowledged. MWCCS data collection is also supported by UL1-TR000004 (UCSF CTSA), P30-AI-050409 (Atlanta CFAR), P30-AI-050410 (UNC CFAR), and P30-AI-027767 (UAB CFAR). VCM received support from the Emory CFAR (P30-AI-050409). The funders had no role in study design, data collection and analysis, decision to publish, or preparation of the manuscript.

**Competing interests:** VCM has received investigator-initiated research grants (to the institution) and consultation fees (both unrelated to the current work) from Eli Lilly, Bayer, Gilead Sciences, and ViiV. The remaining authors declare that they have no competing interests.

## Results

The meta-analysis unveiled a total of 2,021 cell-type unique significant CpG sites for five inferred cell types. Among these inferred cell-type unique CpG sites, the concordance rate in the three cohorts ranged from 96% to 100% in each cell type. Cell-type level meta-EWAS unveiled distinct patterns of HIV-associated differential CpG methylation, where 74% of CpG sites were unique to individual cell types (false discovery rate, FDR <0.05). CD4+ T-cells had the largest number of unique HIV-associated CpG sites (N = 1,624) compared to any other cell type. Genes harboring significant CpG sites are involved in immunity and HIV pathogenesis (e.g. CD4+ T-cells: *NLRC5*, *CX3CR1*, B cells: *IFI44L*, NK cells: *IL12R*, monocytes: *IRF7*), and in oncogenesis (e.g. CD4+ T-cells: *BCL family*, *PRDM16*, monocytes: *PRDM16*, *PDCD1LG2*). HIV-associated CpG sites were enriched among genes involved in HIV pathogenesis and oncogenesis that were enriched among interferon-α and -γ, TNF-α, inflammatory response, and apoptotic pathways.

## Conclusion

Our findings uncovered computationally inferred cell-type specific modifications in the host epigenome for people with HIV that contribute to the growing body of evidence regarding HIV pathogenesis.

## Author summary

The host epigenome (i.e., the DNA methylome) plays a pivotal role in HIV-1 viral integration, maintenance, and activation or silencing. Epigenome-wide association studies have identified many DNA methylation CpG sites in bulk cells associated with HIV infection, which offer an aggregated view of the average DNA methylation across cell types. To understand the effect of DNA methylation on chronic HIV infection at the cell-type level, we applied a computational method to deconvolute DNA methylation from blood bulk cells into five cell types: CD4+ T cell, CD8+ T cell, B cell, Nature Killer cell, and Monocytes and conducted an epigenome-wide analysis for HIV infection in each inferred cell type. Our results show that most of the HIV infection associated CpG sites are unique to each cell type. The largest number of significant CpG sites for HIV infection are from inferred CD4+ T cells. Significant CpG sites are located on genes that are enriched in interferon-α and -γ pathways, which are known to be key players in HIV infection. The findings help us understand the cellular relationship between DNA methylation and HIV infection.

## Introduction

With successful antiretroviral therapy (ART), people with HIV (PWH) have a similar lifespan to the general population [1]. However, the health span of PWH remains 9 years shorter [1] because of a high burden of comorbid chronic diseases such as cardiovascular diseases [2], diabetes, and non-AIDS-related cancers [3]. The prevalence of non-AIDS-related cancers among PWH is significantly higher compared to that among the people without HIV (PWoH) [4, 5], especially in PWH who are not virally suppressed [6,7]. It is important to characterize the

underlying mechanisms involved in HIV pathogenesis that may contribute to increased risk of other comorbid diseases for PWH.

Host epigenetic modifications play critical roles in HIV-1 induced cellular reprogramming at different stages of HIV-1 pathogenesis, including viral integration, maintenance, activation, and silencing [8]. Upon the integration of HIV-1 into the host genome, chromatin in the infected cells undergoes profound reorganization to control the virus by affecting proviral long terminal repeat (LTR) promoter complex formation [9]. HIV-1 proteins (e.g. Tat) in turn change the cellular environment to facilitate virus survival and replication by disrupting chromatin structure and altering gene expression in the host cell. For example, expression of histone methyltransferase (i.e., *DNMT3A*, *DNMT3B*) and histone deacetylase (i.e., *HDAC2*, *HDAC3*) genes are significantly upregulated in host cells infected with HIV-1 [10]. Additionally, although some cells infected with HIV display normal expression of the BCL family of anti-apoptotic proteins permitting apoptosis and viral propagation, other infected cells overexpress these proteins [11], thereby promoting persistence of latently infected cells [12], which poses a barrier to HIV eradication. Such dynamic virus-host genomic interaction results in distinct epigenetic profiles among different immune cell types in response to environmental changes.

Epigenome-wide association studies (EWAS) of the host methylome have identified numerous significant CpG sites for different stages of HIV infection. During the acute stage of HIV infection, as many as 22,697 methylation sites are altered by HIV-1 [13]. ART initiation reverses patterns of DNA methylation in less than 1% of the altered CpG sites, leaving the majority of CpG sites with methylation states persisting into the chronic stage of infection even among virally suppressed individuals [13]. Of note, changes in DNA methylation following ART were only weakly associated with the treatment outcomes such as recovery of CD4+ T-cell counts and CD4+/CD8+ T-cell ratio [14]. Independently replicated studies have identified several CpG sites and genes associated with HIV infection, including three *NLRC5* promoter CpG sites that are less methylated in PWH with or without ART compared to PWoH [15–18]. DNA methylation profiles are predictive of HIV progression, frailty, and mortality for PWH [19–21]. Together, these findings demonstrate the critical role of DNA methylation regulation in HIV pathogenesis and the importance of further characterization of HIV-associated methylation abnormalities among PWH.

While EWAS in whole blood have identified CpG sites such as *NLRC5* that have been replicated by different studies, they provide limited insight into the epigenetic modifications of specific immune cell populations that underlie the pathogenic effects of HIV. HIV-1-induced alterations to the epigenomes of specific immune cell types have not been fully understood. Immune cells that originate from different lineages show distinct DNA methylation patterns [22,23]. Thus, conventional blood- or peripheral blood mononuclear cell (PBMC)-based EWAS may confound the detection of cell-type specific CpG sites for HIV-1 infection. EWAS signals identified from heterogeneous cells are likely to result from consistent CpG methylation signals across cell types or a sufficiently strong cell-type specific signal that exceeds noise from inconsistent methylation of the same CpG in other cell types. However, only a few studies have conducted EWAS in specific cell types to identify HIV-1 associated CpG sites, and then only in a few cell types and evaluated in modest sample sizes [13,24]. For example, one study showed that the number of CpG sites changed by HIV-1 were approximately 100 times greater in monocytes than CD4+ T-cells in the acute stage of HIV infection [13]. Little is known about the impact of HIV-1 infection on the epigenome of other cell types such as B cells, CD8+ T-cells, natural killer (NK) cells, or granulocytes, in addition to monocytes and CD4+ T-cells. DNA methylation in specific cell types regulates the genes responding to, but is ultimately altered by, HIV-1 infection [25,26].

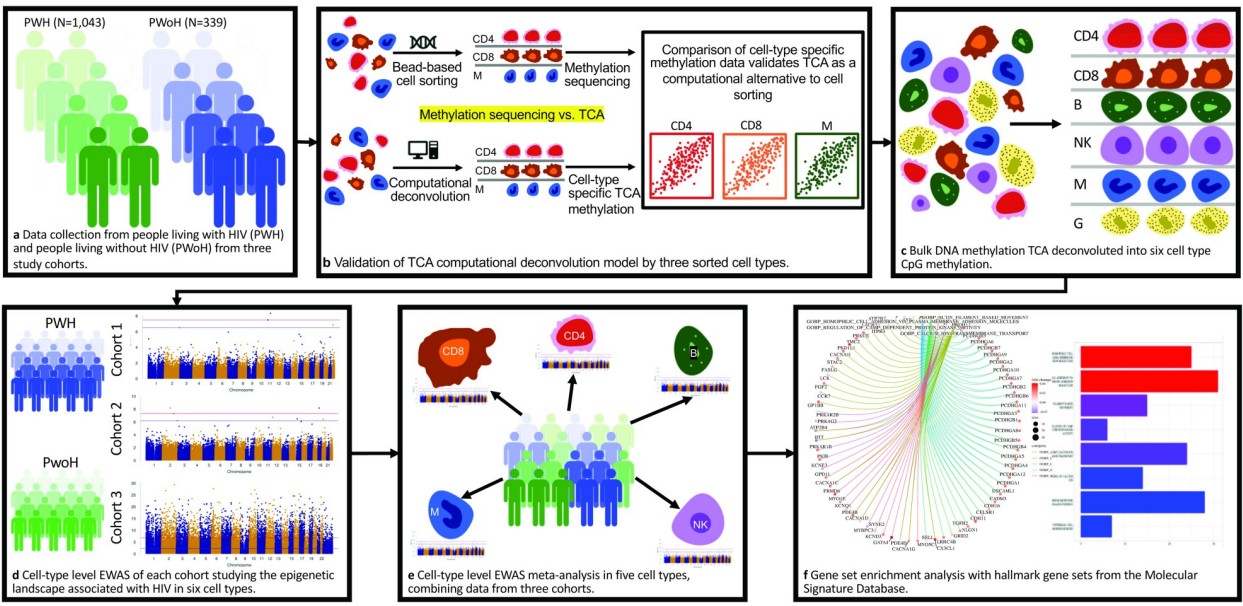

**Fig 1. Flowchart of analytical strategies.** PWH: People with HIV; PWoH: People without HIV. TCA: Tensor Component Analysis; EWAS: Epigenome-wide Association Study. CD4: CD4+ T-cells; CD8: CD8+ T-cells; B: B cells; NK: Natural Killer cells; M: Monocytes; G: Granulocytes.

Cell-type specific profiling of DNA methylomes in large population samples is technically challenging and cost prohibitive, especially for less abundant cell types in blood. Recently, computational methods have been developed and applied to deconvolute cell-type specific methylation signals from bulk PBMCs or whole blood samples [27–32]. You *et al.* (2021) successfully dissected several well-known smoking-associated hypomethylation signatures that derived from myeloid lineage immune cells [33].

In this study, we hypothesized that alterations in CpG methylation in the genome of PWH differ among immune cell types and that HIV-associated CpGs are enriched among genes involved in HIV pathogenesis, but potentially other comorbid conditions including cancer development and progression. We applied a tensor composition analysis (TCA) method to computationally deconvolute cell-type specific methylation data in PBMCs without sorting cells [34]. We first validated the TCA deconvoluted methylomes by direct bisulfite DNA sequencing of sorted CD4+ T-cells, CD8+ T-cells, and CD16+ monocytes from the same PBMC specimen for a subset of the sample. We then deconvoluted methylation data from whole blood or PBMCs into CD4+ T-cells, CD8+ T-cells, B cells, NK cells, and monocytes in 1,382 samples. These cell-type specific EWAS for HIV infection were conducted in three cohorts: the Veteran Aging Cohort Study (VACS) [35] for men (N = 718), the Women's Interagency HIV Study (WIHS) [36] for women (N = 436), and a public dataset GSE217633 including both men and women (N = 228) (S1 Table). To the best of our knowledge, this is the first and largest cell-type specific EWAS for HIV infection. A flowchart of analytical strategies is presented in Fig 1.

## Methods

### Ethics approval and consent to participate

The study was approved by the committee of the Human Research Subject Protection at Yale University and the Institutional Research Board Committee of the VA Connecticut Healthcare

System, and by the participating clinical sites. All subjects provided written informed consent. All methods were carried out in accordance with relevant guidelines and regulations, including those of the Declaration of Helsinki. The views and opinions expressed in this manuscript are those of the authors and do not necessarily represent those of the Department of Veterans Affairs or the United States government.

## Sample characteristics and DNA methylation profiling

**Cohort 1 (VACS).** The VACS is a nationwide longitudinal cohort of veterans including PWH and PWoH designed to study HIV infection and disease progression. The methylation data from this cohort, profiled using 450K BeadChips, were previously published and deposited in GSE107080 and GSE107082 in the Gene Expression Omnibus (GEO) repository. A total of 718 samples (PWH = 614; PWoH = 104) from the VACS Biomarker Cohort, a subset of the VACS, were included in the analysis (S1 Text). The majority (86%) of the VACS sample were of African American/Black (AA) ancestry and all samples were collected from male participants. PWoH were slightly older (i.e., an average of 1.6 years greater) compared to PWH (p = 0.05). Among PWH, 78.3% of the participants had good adherence of ART, the majority were virally suppressed (defined as an undetectable HIV RNA viral load by commercial test), and an average log10 HIV RNA viral load of 2.7 ± 1.2 among the unsuppressed PWH. Approximately half of the participants were self-reported cigarette smokers and the majority reported drinking alcohol. These demographic and clinical variables were adjusted for in the EWAS model.

**Cohort 2 (WIHS).** Clinical data and specimens used in this study were collected by the Women's Interagency HIV Study (WIHS), now the Multicenter AIDS Cohort Study (MACS)/ WIHS Combined Cohort Study (MWCCS) [37] (S1 Text). MWCCS is the largest observational cohort of HIV infection in the United States. WIHS included 245 samples from PWH and 191 PWoH from diverse ancestral populations and all samples were collected from female participants. PWH were an average of 5.2 years older compared to PWoH (p = 2.0E-07). Among PWH, 82.6% of the participants reported good ART adherence, the majority had undetectable HIV RNA viral load, and an average log10 HIV RNA viral load of 2.1 ± 0.6 among unsuppressed PWH. Among PWoH, there were more self-reported smokers (75.4%) (p = 0.0027) and participants who drank alcohol (51.8%) (p = 7.9E-05) compared to PWH. Those variables were adjusted for in the EWAS model. Demographic and clinical characteristics are presented in S1 Table.

**Cohort 3 (GSE217633) [14].** We leveraged a publicly available dataset to provide a replication cohort, followed by a meta-EWAS of the three cohorts. The dataset included 187 participants from the NEAT001/ANRS143 clinical trial: a randomized open-label, non-inferiority trial conducted in 78 clinical sites in 15 European countries for non-inferiority over 96 weeks of ritonavir-boosted darunavir plus raltegravir versus ritonavir-boosted darunavir combined with tenofovir disoproxil fumarate and emtricitabine. Forty-four additional age and sex matched PWoH were included as a control group [14]. In this analysis, we used a single timepoint with DNAm data from PWH pre-ART treatment and PWoH for a total of 228 participants for the EWAS on HIV infection. The cohort was predominantly men (88.2%) and the majority were of European ancestry (83.1%). The average pre-ART HIV RNA viral load (log10) was 4.7 ± 0.5.

Methylation of DNA extracted from whole blood in cohort 1 was profiled using the Illumina HumanMethylation 450K Beadchip. Similarly, methylation of DNA from PBMCs in the cohort 2 and from whole blood in the cohort 3 was profiled using the Illumina HumanMethylation EPIC Beadchip. Only 408,366 CpG sites that were common to both the 450K and EPIC

arrays were used in three cohorts. Among the common CpG sites between two arrays, CpG sites with a variance greater than 0.002 were retained for application of TCA. More information about DNA methylation quality control and deconvolution are presented below in the Methods and in S1 Text.

## Deconvolution of DNA methylation from bulk cells to individual cell types

To deconvolute bulk methylation at each CpG site to specific cell types, TCA requires a DNA methylation data matrix in heterogeneous cells and cell type proportions for each sample in the cohort. We first estimated the proportion of six cell types from the DNA methylation data for each sample using the Houseman method through the GLINT (version 1.0.4) script. Six cell type proportions from whole blood were estimated for cohorts 1 and 3 (CD4+ T-cells, CD8+ T-cells, B cells, NK cells, monocytes, granulocytes). Because isolating PBMCs, which were the source for DNA methylation analysis in cohort 2, results in the near-total depletion of granulocytes, five cell type proportions without granulocytes were estimated in cohort 2. The estimated proportions of each cell type in each cohort are presented in S1 Fig.

One of the advantages of the TCA package is that it estimates the methylation value at a given CpG site for EWAS. While the TCA R package also provides the "tcareg" function, which performs linear regression with covariate adjustment without the production of cell type methylomes [34], this function does not support a comprehensive EWAS approach to further control for unknown confounding effects. Thus, we adapted the TCA function for deconvoluting methylation level for each CpG to enable more robust adjustment for confounders for downstream EWAS. The CpG sites from bulk cells were deconvoluted to either six cell types using a whole blood methylation matrix (cohorts 1 and 3) or five cell types from a PBMC methylation matrix (cohort 2). To assess whether TCA removed the cell-type proportion confounding effects, we performed correlation analysis between 30 Principal Components (PCs) derived from TCA-deconvoluted methylation in each cell type and cell type proportion in cohorts 1 and 2, for which the methylation intensity for each probe was available to us. Robust deconvolution of cell-type specific methylation from bulk DNA methylation data would be expected to be reflected by little correlation of PCs with cell type proportion. Our results showed that correlations between PCs and cell proportion were strong in bulk cells but weak in each cell type following the deconvolution, suggesting that TCA robustly deconvoluted methylation to individual cell types, removing cell type confronting effects in these two cohorts (S2 Fig).

## Capture methylation sequencing

Three cell types, CD4+ T-cells, CD8+ T-cells, and CD16+ monocytes, were isolated from 4 PBMC samples using a magnetic bead-based method [38]. DNA was extracted from each isolated cell type. Methylation sequencing target enrichment library preparation was performed per manufacturer protocol (Agilent). Samples were sequenced using 100bp paired-end sequencing on an Illumina HiSeq NovaSeq according to Illumina standard protocol. Detailed quality control and data processes are presented in S1 Text. CpG sites were annotated using Homer annotatePeaks.pl, including intergenic, 5'UTR, promoter, exon, intron, 3'UTR, transcription start site (TSS), and non-coding categories. CpG island, shore, shelf, and open sea annotation was defined by locally developed bash and R scripts based on genomic coordinates (hg19) of CpG islands from the UCSC genome browser. CpG shore was defined as up to 2 kb from CpG islands and CpG shelf was defined as up to 2 kb from a CpG shore. Methylation CpG sites on the X and Y chromosomes were removed for subsequential analyses.

## Benchmarking deconvoluted DNA methylome data using capture bisulfite sequencing in CD4+ T-cells, CD8+ T-cells, and monocytes

We validated the accuracy of the TCA-derived estimates of DNA methylation at each CpG site by comparing the methylation β-value for the top 10,000 most-variable CpG sites between the TCA-deconvoluted and the directly measured methylation β-value for CD4+ T-cells, CD8+ T-cells, and CD16+ monocytes. We selected the top 10,000 most variable CpG sites among the samples for this comparison, which was performed using Pearson correlation analysis with significance set at p<0.05. Correlation coefficients for each pair were 0.96 in CD4+ T-cells, 0.97 in CD8+ T-cells, and 0.96 in CD16+ monocytes (S3 Fig). The distributions of CpG methylation in each cell type derived by the TCA and MC-seq methods were almost identical, suggesting that TCA is a robust and effective deconvolution method.

## Cell-type based epigenome-wide association analysis

We performed a cell-type based EWAS for HIV-infection using TCA-deconvoluted methylation beta values for each cell type. In each cell type, we adapted the incorporation of the Control Probe Adjustment and reduction of global CORrelation (CPACOR) pipeline proposed by Lehne *et al.* [39] to provide further quality control and to remove batch effects. CPACOR leverages control probes that are designed in 450K and EPIC arrays to adjust for background noise. Principle Components 1–30 derived from control probes were adjusted for in the model per the CPACOR recommendation. A two-step regression analysis was conducted following the pipeline. The first regression model addressed global covariates that may confound the association of methylation with HIV infection. Although TCA significantly reduced the confounding effects of cell type proportion (S2 Fig), we included six cell type proportions as additional covariates in the model to further remove their residual effect. We first estimated the residual β using regression model (1):

$$\beta\ value\ (quantile\ normalized, QN) \sim age + race + smoking\ status + White\ Blood\ Cell\ (WBC)$$
$$+ 6\ cell\ type\ proportions + PCs\ 1\text{-}30\ on\ intensities\ of\ control\ probe$$

We then performed a principal component analysis on the resulting regression residual β values and regressed out the first 5 PCs to further control for unmeasured confounders in the regression model (2). The 5 PCs adjustment resulted in low inflation.

$$\beta\ value(QN) \sim age + race + smoking\ status + WBC + 6\ cell\ type\ proportions$$
$$+ PCs\ 1\text{-}30\ on\ intensities\ of\ control\ probes + PCs\ 1\text{-}5\ on\ residuals\ from\ model \qquad (1)$$

Significance was set at a false discovery rate (FDR) <0.05. To further confirm the correction of global confounders, we performed Pearson correlation analysis between the first 30 PCs on residual methylation from model (1) and array batch, demographic, clinical, and cell type confounders. The cutoff for correlation analysis was set at p<0.05.

## Correlation analysis of HIV-associated CpG sites between the three cohorts

In each cell type, we selected CpG sites with FDR<0.05 in cohort 1 for multivariable correlation analysis among three cohorts. Correlation of the effect sizes at each resulting CpG between the three cohorts was performed by the corr.test function in the pych R package. The average r and p values were calculated for each cell type. The significance threshold was set at p<0.05. We also compared the direction of effect of each CpG among the three cohorts.

## Cell type-based EWAS meta-analysis

We conducted an EWAS meta-analysis for each cell type based on the summary statistics from the three cohorts. We excluded granulocytes from the meta-analyses to provide a consistent set of computationally inferred cell-type specific epigenome profiles shared among the three cohorts. Effect sizes and p-values for each probe were obtained from analyses in each cohort. We performed inverse-variance weighted meta-analysis, with scheme parameters of sample size and standard error as implemented in the METAL program [40], combining summary statistics from the three sample sets. We investigated heterogeneity between the three samples using the $I^2$-statistic. CpG sites with $I^2 > 50\%$ were excluded from subsequent analysis.

## Gene set enrichment analysis

Genes adjacent to CpG sites in meta-EWAS for each cell type were selected for gene set enrichment analysis. The cutoff of FDR<0.1 was used to ensure a sufficient number of genes selected for enrichment analysis. We focused on the hallmark gene sets from the Molecular Signature Database (https://www.gsea-msigdb.org/gsea/msigdb/). Enrichment analyses using GO annotations were also performed.

## Results

### Cell type-based EWAS identified differentially methylated positions (DMPs) for HIV infection in men with HIV: Cohort 1 (VACS)

In cohort 1, we identified 496 epigenome-wide significant (EWS) DMPs associated with HIV infection in bulk DNA methylome data from whole blood (FDR<0.05) (Figs 2A and S4A and S2 Table). The significant DMPs included those previously reported by us and other groups. Examples include two previously replicated associations at *NLRC5* (cg16411857, t = -9.42, FDR = 3.58E-14 and cg07839457, t = -9.05, FDR = 5.68E-10) [15,16,41]. Hypomethylation of *HCP5* that was previously linked to HIV infection [16] also reached EWS in this study (cg18808777, t = 5.61, FDR = 7.81E-04). While replication of these well-validated CpG sites and genes is important, whether these DMPs originate from specific cell types or sub-groups of cell types is unknown.

At the cell-type level, we identified considerably more EWS DMPs across the six cell types than in whole blood: 2,163 in CD4+ T-cells, 106 in CD8+ T-cells, 8 in B cells, 317 in NK cells, 21 in monocytes, and 697 in granulocytes (Figs 2A and S4B and S3–S8 Tables). The majority of DMPs in each cell type differed among the six cell types; a small number of DMPs were common in more than one cell type. For example, one DMP, *LPCAT1* cg16272981, was significant in four cell types (CD4+ T-cells: FDR = 1.15E-07; B cells: FDR = 0.0016; NK cells: FDR = 0.005; monocytes: FDR = 3.29E-06, and granulocytes G: FDR = 6.94E-10).

In CD4+ T-cells, we found more hypomethylated (N = 1,336) than hypermethylated (N = 827) CpG sites in samples from PWH relative to PWoH. The 2,163 DMPs were located in 1,407 genes. The HIV-associated loci previously reported in CD4+ T-cells were replicated in this cohort. *LPCAT1* cg16272981 showed the largest effect (16.1% less methylated in samples from PWH relative to PWoH). The 30 top ranked DMPs were located in 14 genes (i.e. *LPCAT1*, *SLC17A9*, *RUNX3*, *KLF7*, *SEPT9*, *PEX14*, *NLRC5*, *SPOCK2*, *SPATAS*, *MYT1L*, *CAPN11*, *SEMA3G*, *BCL9*, *XYLT1*) (FDR = 8.49–09~ 5.21E-06). Some genes harbored multiple DMPs. For example, 4 EWS DMPs were located in *RUNX3*, a well-recognized tumor suppressor of gastric, colon and many other forms of solid tumors [42]. The majority of HIV-associated DMPs in CD8+ T-cells were hypomethylated. Six out of 8 DMPs in B cells were hypomethylated. *LPCAT1* cg16272981, significantly enriched in CD4+ T-cells, showed the

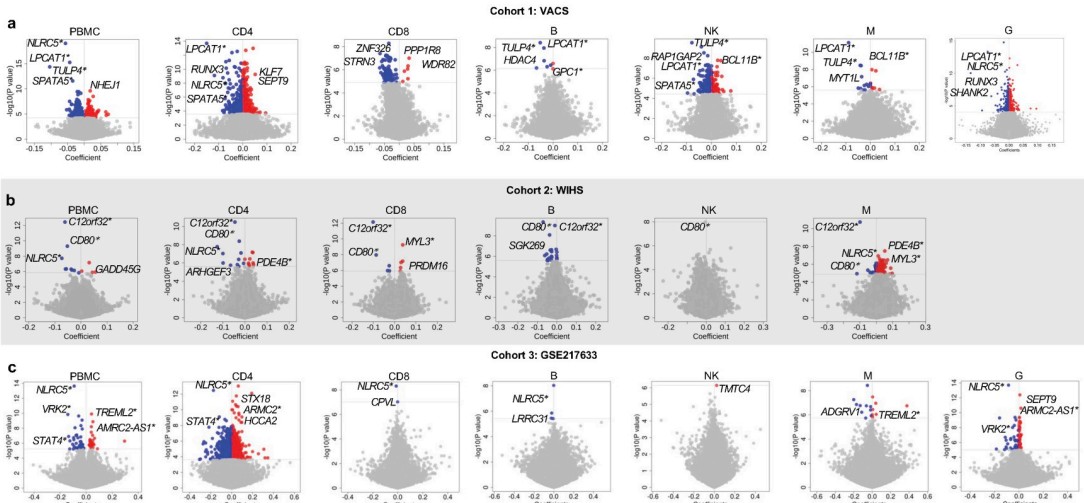

**Fig 2. Summary of cell-type level EWAS in the three cohorts in each cell types (CD4+ T-cells, CD8+ T-cells, B, Natural Killer, Monocyte, Granulocytes).** (A) Cohort 1: Volcano plots for the VACS cohorts with top common and unique hyper- and hypomethylated gene-associated sites annotated, where DNA was derived from whole blood samples. (B) Cohort 2: Volcano plots for the WIHS cohort with similar annotations. Granulocyte type is excluded. (C) Cohort 3: Volcano plots for the data from GSE217633. PBMC: peripheral blood mononuclear cell; EWAS: Epigenome-wide Association Study; VACS: Veteran Aging Cohort Study; WIHS: Women's Interagency HIV Study; DMPs: Differential Methylation Positions. *Significant genes shared between at least two cell types.

strongest EWS association in B cells (t = -5.97, FDR = 0.003). Other significant DMPs were located on *TULP4*, *ETS1*, *KCNK9*, *STAT3*, *HDAC4*, and *GPC1*. Of note, 6 out of 8 DMPs were common between CD4+ T-cells and B cells except for *ETS1* and *GPC1*. In NK cells, 159 DMP sites overlapped with other cell types. *TULP4* cg02571055 was hypomethylated in B cells and the strongest EWS association in NK cells (t = -6.40, FDR = 0.0001). Among 21 significant CpG sites in monocytes, the most significant DMP was *LPCAT1* cg16272981 (t = -6.97, FDR = 3.29E-06), followed by *TULP4* cg02571055 (t = -5.99, FDR = 4.94E-04), which was also identified in B cells and NK cells. Thirteen DMPs were observed in monocytes, including DMPs located in *NOTCH4* and *IGSF9*. In granulocytes, two DMPs in *LPCAT1* (cg16272981and cg08697251) and three DMPs in *NLRC5* (cg16411857, cg07839457, and cg05757530) were among the top ranked significant DMPs for HIV infection.

## Cell-type EWAS for HIV infection in women with HIV: Cohort 2 (WIHS)

The EWAS of women with HIV using PBMCs was carried out by applying the same regression model and adjusting for the same covariates as for the EWAS in the VACS. We identified 13 EWS DMPs in PBMCs associated with HIV infection (Figs 2B and S4C and S9 Table). Consistent with the VACS sample, *NLRC5* cg07839457 was one of the significant CpG sites (t = -5.76, p = 1.85E-08). *NLRC5* cg16411857 showed near epigenome-wide significance (t = -4.51, p = 8.95E-06). Other EWS DMPs were located in *C12orf32*, *CD80*, *GADD45G*, *TXNIP*, *TMEM49*, *SGK269*, *DUSP16*, *RAC2*, *TNIP3*, and *GLB1L2*. In addition, we performed a second EWAS in PBMCs using all ~870K CpG sites in the EPIC array. We identified 47 significant CpG sites associated with HIV infection (FDR<0.05). Compared to the result from the EWAS including only common CpG sites in the 450K array, we found 43 more significant CpG sites. (S5 Fig and S10 Table). The small number of significant CpG sites in this analysis is likely due to a stringent analytical approach to adjust known and unknow confounding factors with HIV

infection (see Methods). Importantly, all previously 13 significant CpG sites were also significant in the EPIC-EWAS with slightly increased p values. For example, the FDR value for the top CpG cg12051710 in *C12orf32* was 1,47E-07 in 450K-EWAS but was 3.03E-07 in EPIC-EWAS.

For the cell-type level EWAS, we identified 153 significant DMPs among the 5 cell types: 20 for CD4+ T-cells, 10 for CD8+ T-cells, 22 for B cells, 1 for NK cells, and 100 for monocytes (all FDR<0.05) (Figs 2B and S4D and S11–S15 Tables). Several DMPs are worthy of mention. *C12orf32* cg12051710 displayed the strongest association in four out of five cell types: CD4+ T-cells (t = -6.85; FDR = 1.35E-05), CD8+ T-cells (t = -7.45, FDR = 2.95E-07), B cells (t = -6.28, FDR = 2.02E-04), and monocytes (t = -6.94 p = 7.43E-06). *CD80* cg13458803 was hypomethylated in multiple cell types: CD8+ T-cells (t = -5.86, FDR = 0.0001), B cells (t = -6.41, FDR = 0.0002), and NK cells (t = -5.86, FDR = 0.004). Of note, *NLRC5* cg07839457 was one of the top ranked DMPs in CD4+ T-cells in the WIHS (t = -5.78, FDR = 0.00537), consistent with the EWAS in the VACS.

## Replication of cell-type EWAS for HIV infection using publicly available data: Cohort 3

PWH in cohorts 1 and 2 were on ART, which may confound the effect of HIV-1 for HIV infection. To replicate the results for HIV-1 infection in the absence of ART, we conducted the whole blood and cell-type based EWAS in cohort 3 between pre-ART PWH and PWoH. In this smaller sample cohort, we found that 75 CpG sites in whole blood were differentially methylated between PWH before ART and PWoH (Figs 2C and S4E and S16 Table). Hypomethylation of cg16411857 and cg07839457 on *NLRC5* was consistent with cohorts 1 and 2. Of note, among the 75 significant CpG sites, 39 CpG sites including the top ranked DMPs were identical as the previous report in this cohort. At the gene level, 28 out of 49 genes were the same as the original paper [14]. Furthermore, leveraging the available DNA methylation data of pre-ART and post-ART in this cohort, we performed an EWAS on ART and identified 807 significant CpG sites (FDR <0.05) (S6 Fig and S17 Table), which included 183 identical CpG sites with the previous study [14].

Cell-type based EWAS revealed 2,652 EWS DMPs in CD4+ T cells, 2 DMPs in CD8+ T cells, (i.e. cg16411857 and cg07774680), 4 DMPs in B cells (i.e. cg16411857, cg10217889, cg22158051, cg24880620), 1 DMP in NK cells (i.e. cg19349476), 17 DMPs in monocytes, and 111 DMPs in granulocytes (Figs 2C and S4F and S18–S23 Tables). Given our purpose of replicating the results from cohorts 1 and 2, we set a liberal p-value cutoff of p<0.05. We found that a large number of significant CpG sites/genes in cohort 1 were replicated in cohort 3. For example, in CD4+ T cells, 86% of genes harboring significant CpGs were the same between cohorts 1 and 3, including *NLRC5, SLC17A9, CCL16, TULP4, SHANK2*. In monocytes, 17 of 18 genes harboring significant CpG sites in cohort 1 were replicated in cohort 3 including examples of *TULP4, BCL11B, PDCD1LG2*. Therefore, TCA-deconvoluted cell-type EWAS enables identification of DMPs for HIV infection across distinct cohorts.

## Concordance of DMPs for HIV infection in the three cohorts

The distinct demographic and clinical characteristics of the three cohorts could influence the pattern of HIV-associated DMPs identified in each cohort, their comparability across the cohorts, and therefore the generalizability of the findings to other studies. To address this possibility, we conducted a correlation analysis of effect sizes for each DMP across data from bulk samples (whole blood, PBMC) and each cell type among three cohorts. We found that effect sizes of the same CpG site among the three cohorts were well correlated in bulk cells (average

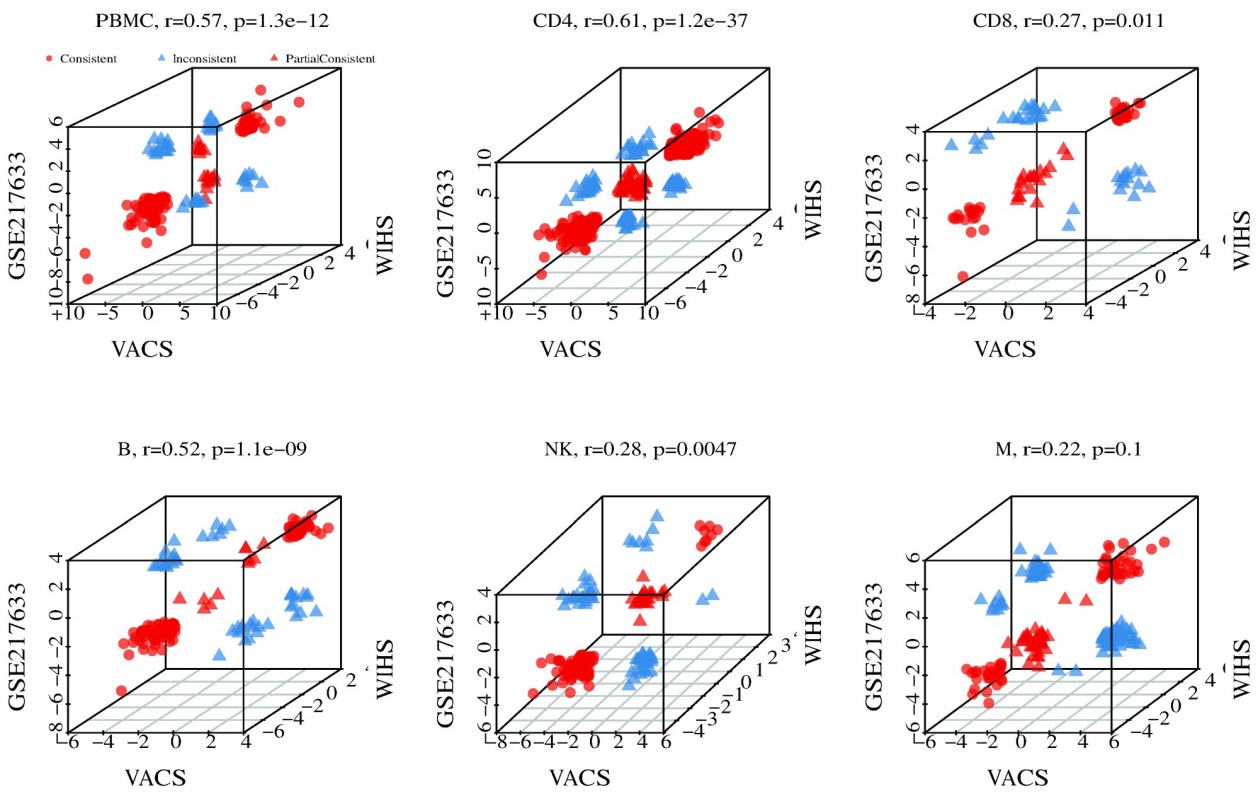

**Fig 3. Correlation of effect sizes in PBMC and each cell type among three cohorts.** Red dots: consistent direction among three cohorts; red triangle: consistent direction between cohort 1 (VACS) and cohort 3 (GSE217633), which are both whole blood based methylation data. Blue triangle: inconsistent direction among three cohorts. PBMC: Peripheral blood mononuclear cells; VACS: Veteran Aging Cohort Study; WIHS: Women's Interagency HIV Study.

r = 0.57, p = 1.3E-12) and in four out of five cell types (Fig 3). At the cell type level, the correlation coefficients of DMPs were strongest in CD4+ T-cells (average r = 0.61, p = 1.2E-37), followed by B cells (r = 0.52, p = 1.10E-09), NK cells (r = 0.28, p = 0.0047), and CD8+ T-cells (r = 0.27, p = 0.01). Correlation of DMPs in monocytes was not significant (r = 0.22, p = 0.1) (Fig 3). The directions of the correlations for the majority of DMPs in PBMCs and CD4+ T-cells were concordant among the three cohorts: 91/157(58%) in PBMCs and 276/413 (67%) in CD4+ T-cells. Correlation of DMPs in whole blood between two cohorts 1 and 3 was greater than the correlation between cohort 2 (PBMCs) and cohort 3 (whole blood) (Fig 3). The correlation analysis results underscored the value of EWAS meta-analysis of the three cohorts.

## EWAS meta-analysis by cell type identified common and specific DMPs for HIV infection

Meta-EWAS in bulk cells revealed 724 DMPs, including top significant genes *NLRC5*, *LPCAT1*, *BCL2L*, *SHANK2*, *SLC17A9*, *STAT4*, *TULP4*, *HCP5*, and *PSMB8* (Fig 4A and S24 Table). Meta-EWAS by cell type identified 2,504 epigenome-wide DMPs in CD4+ T-cells, 18 DMPs in CD8+ T-cells, 306 DMPs in B cells, 377 DMPs in NK cells, and 286 DMPs in monocytes (Fig 4B and 4C and S25–S29 Tables). Several genomic regions harbored DMPs that were common to more than one cell type. Multiple loci on different chromosome regions were common in more than one cell type, such as chromosome 3 (*ARHGEF3*, *DNAJB8*, *CCRL*,

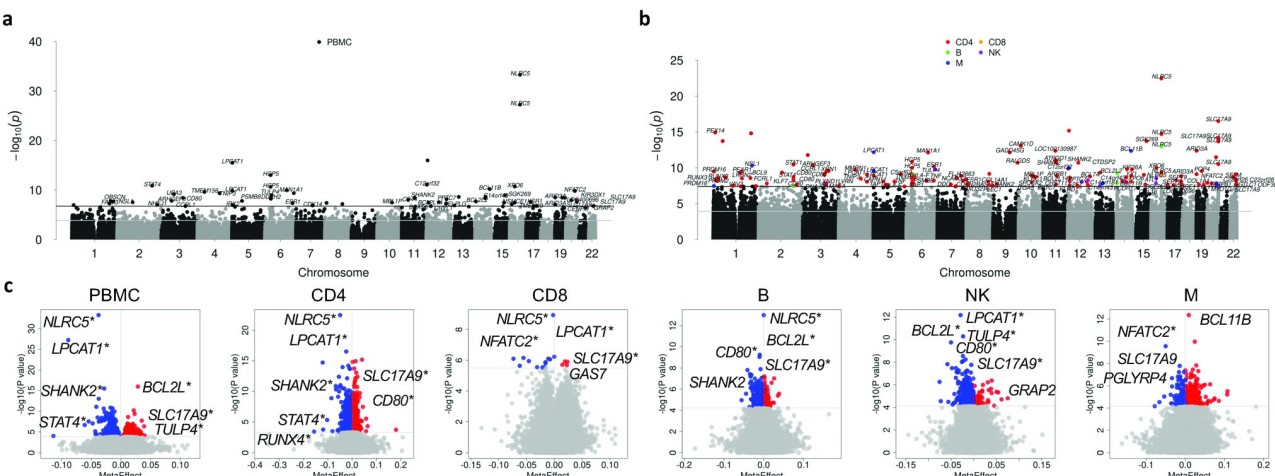

**Fig 4. Summary of cell-type level epigenome-wide meta-analysis of the combined three cohorts: Veteran Aging Cohort Study (VACS), Women's Interagency HIV study (WIHS) data, and GSE217633.** (A) Manhattan plot of epigenome-wide significant CpG sites in bulk cells prior to computational deconvolution of data into cell-type-specific methylation. (B) Manhattan plot of epigenome-wide significant CpG sites after computational deconvolution into cell-type-specific signals. (C) Volcano plots of hyper- and hypomethylated DMPs between PWH and PWoH in each cell type following Meta-EWAS (Epigenome-wide Association Study) for five cell types (CD4+ T-cells, CD8+ T-cells, B cells, Natural Killer cells, and Monocytes). DMP: Differential Methylation Position. *Significant genes shared between at least two cell types.

*CD80*), chromosome 5 (*LPCAT1*), chromosome 10 (*RUNX2*), chromosome 11 (*SHANK2*), chromosome 16 (*NLRC5*), and chromosome 20 (*SLC17A9*) ([Fig 5A]). Several top ranked CpG sites were located in genes encoding for transcription factors. For example, in meta-EWAS of CD4+ T cell, cg00676801 on *STAT1* was hypomethylated in HIV-infection (meta-FDR: 3.39E-

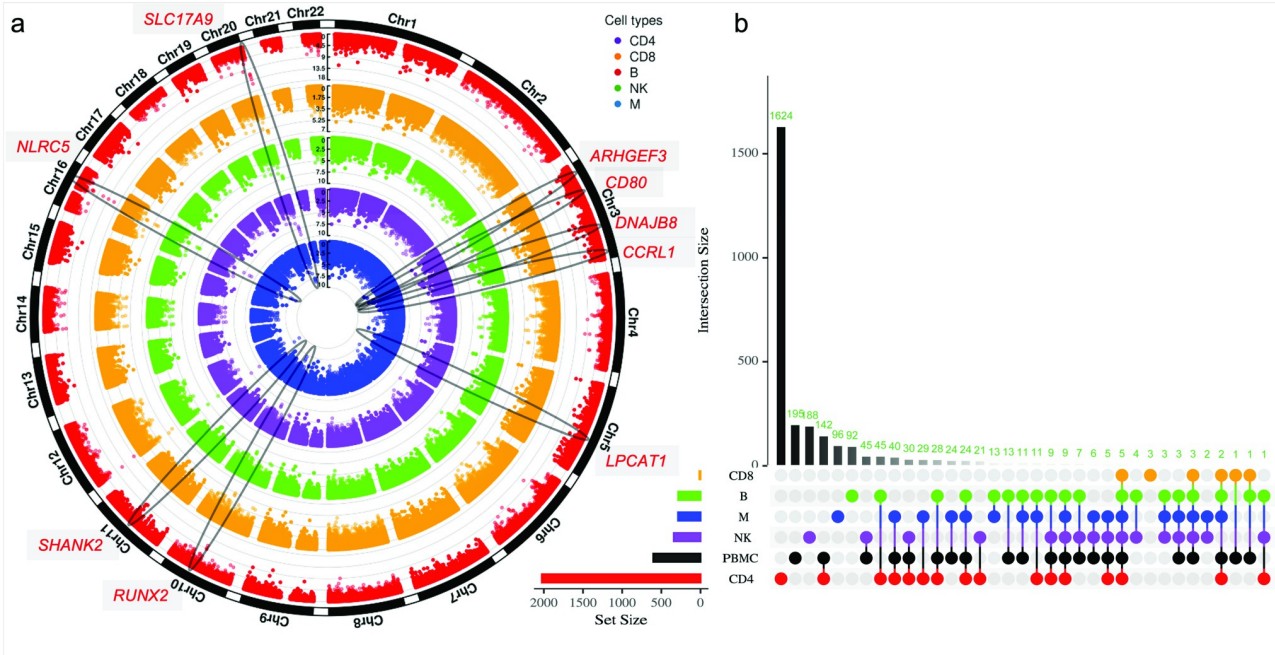

**Fig 5. Common and distinct DMP profiles among cell types following cell-type level meta-EWAS.** (A) Circos plot displaying the landscape of DMPs in each cell type and the regions shared HIV-associated DMPs among five cell types. (B) Unique and common DMPs among five cell types and peripheral blood mononuclear cells. Intersection size represents the number of shared DMPs between the designated cell types, as designated by conjoined bubbles below. DMP: Differential Methylation Position.

07). *STAT1* is a transcription factor involved in upregulating genes due to a signal by interferons, cytokines, and growth factors. cg05573412 on *PRDM16*, encoding a zinc finger transcription factor that plays a role in the regulation of developmental processes, was hypermethylated (meta-FDR: 2.06E-06). cg15498134 on *RUNX3*, a gene involved in lineage-specific gene expression in a variety of cells, was hypomethylated in HIV infected samples (meta-FDR: 9.74E-06).

Overall, the majority of DMPs (74%) from the meta-EWAS were unique to each cell type (Fig 5B). Cell-type specific DMPs accounted for 80.2% of DMPs in CD4+ T-cells, 16.7% in CD8+ T-cells, 32.4% in B cells, 55.1% in NK cells, and 33.6% in monocytes. Among lymphocytes, only 1.9% of DMPs overlapped between CD4+ T-cells and B cells, and 1.3% of DMPs between CD4+ T-cells and NK cells. The results suggest that meta-EWAS revealed distinct DNA methylation modifications for HIV infection between cell types. Annotation of significant CpGs showed that the majority of CpG sites were located in gene bodies in each cell type. Regional enrichment analysis showed that DMPs in the promoter region were significant in CD4+ T cells ($\chi^2$ = 5.01; p = 0.025), but not significant in other cell types. The proportion of DMPs in CpG islands was also greater in CD4+ T-cells (S7 Fig).

To better interpret the biological significance of the significant DMPs, as examples, we present the methylation β value for meta-EWAS significant CpG sites differing at least 5% between HIV+ and HIV- in two out of three cohorts in each cell type (S8 Fig).

## Cell-type specific DMPs from meta-EWAS for HIV infection are involved in oncogenesis

Previous studies have demonstrated that pathogen-induced epigenetic alterations cumulatively contribute to cancer development [43], including for HIV-infection. For example, in CD4+ T-cells, *BCL9* for B-cell acute lymphoblastic leukemia (B-ALL) [44]; *GAS7* and *PRDM16* for Acute myelogenous leukemia (AML) [45,46]; *ESR1* for breast cancer [47]; *and GRIN2A* for colorectal, lung, and gastric carcinoma [48] were differentially methylated between PWH and PWoH. In B cells, *PRDM16* and *DNMT3A* for AML, *PDCD1LG2* for Hodgkin's lymphoma, *LMNA* for spritzed tumor, and *BCL11B* for T-cell acute lymphoblastic leukemia (T-ALL) harbored DMPs for HIV infection. Genes in NK cells (i.e., *BCL11B* for T-ALL, *MAP3K7IP2* for prostate cancer) and in monocytes (i.e., *LCK* and *BCL11B* for T-ALL, *PDCD1LG2* for Hodgkin's lymphoma, *PRDM16* for AML, *CACNA1D* for prostate cancer) were found significant in our meta-EWAS.

## HIV-associated DMPs are enriched in gene sets for immunity and cancer biology

Among the set of hallmark genes in the Molecular Signature Database, we found 7 significant pathways in CD4+ T-cells, 2 in B cells, and 4 in NK cells, and 5 in monocytes (FDR<0.05) (S30 Table). No significant hallmark gene pathways were significant in CD8+ T-cells. Multiple pathways were identified that were involved in immune function and cancer biology. In CD4 + T-cells, 4 out of 7 pathways were involved in immunity (i.e., allograft rejection, interferon-γ, inflammatory response, and TNF-α signaling via NFKB). Fig 6A presents the top five ranked pathways in CD4+ T-cells. Multiple genes contained HIV-associated DMPs in the allograft rejection pathway, including hypomethylated DMPs in *CD80*, *CD8A*, *CRTAM*, *HLA-DOA*, *PTPRC*, *STAT1*, *STAT4*, and *CD96* and hypermethylated DMPs in *MBL2* and *CD4*. The allograft rejection pathway and the Interferon-γ response pathway are connected by HIV-associated DMPs located in the following genes: *STAT4*, *STAT1*, *IRF4*, *IRF8*, *TAP1*, *HLA-A*, *HLA-G* and *FAS* (Fig 6B). The above-mentioned pathways are also involved in cancer development

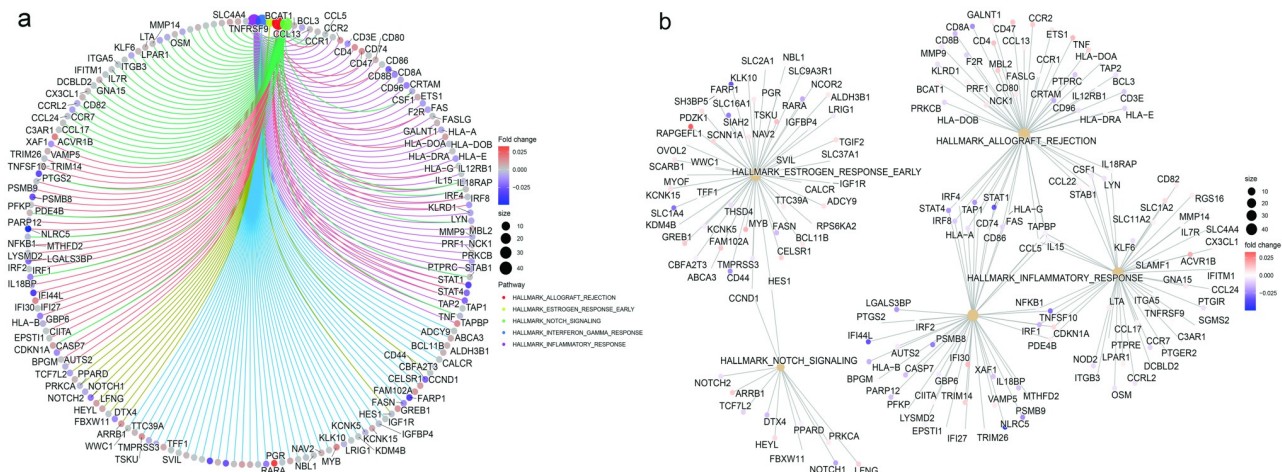

**Fig 6. Gene enrichment analysis of DMPs in the genes enriched on hallmark gene pathways in CD4+ T-cells.** Notably, the allograft rejection, early estrogen response, interferon alpha response, interferon gamma response pathways were significantly enriched. (A) Circos plot showing DMPs in four pathways in CD4+ T cells; (B) Relationships of four significant pathways. DMP: Differential Methylation Position.

and progression. Interestingly, the estrogen response pathway was significant in CD4+ T-cells, B cells, and NK cells. The apoptosis pathway was significant in NK cells. Significant pathways in monocytes included the functions in inflammation (i.e., inflammatory-α, interferon-α) and cancer biology (i.e., pancreatic β cells and hedgehog signaling pathways). The results further underscore the striking enrichment of genes in cell types that feature DMPs for HIV-pathogenesis.

The cell-type based GSEA using meta-significant DMPs were also performed in Gene Ontology (GO), KEGG: Kyoto Encyclopedia of Genes and Genomes, and Wiki pathways. S9 Fig shows top 15 GO, KEGG, and Wiki pathways in each cell type. Overall, immune response, autoimmune disease, and cancer pathways are among top 15 pathways, which corroborate with the results in hallmark pathways.

## Discussion

We identified DMPs for chronic HIV infection in six major immune cell types: CD4+ T-cells, CD8+ T-cells, B cells, NK cells, monocytes, and granulocytes from whole blood samples in two cohorts, and in five cell types (without granulocytes) from PBMCs in one cohort. These include a number of previously reported DMPs associated with HIV infection. Despite differences in demographic and clinical characteristics among three cohorts studied, we found a number of replicated and highly concordant DMPs among cohorts. The majority of DMPs identified in individual cell-type meta-EWAS were unique to each cell type (74%). The occurrence of distinct profiles of HIV-associated DMPs among immune cell types highlights the importance of examining differences in DNA methylation profiles between individual cell types. Among the five cell types from meta-EWAS, the number of DMPs identified in CD4 + T-cells were three-to-ten-fold greater than the other four cell types, suggesting that epigenetic alteration in CD4+ T-cells plays a major role in chronic HIV-infection. More importantly, we found that genes that harbored HIV-associated DMPs are also involved in cancer biology. The identified genes were enriched among hallmark pathways of HIV pathogenesis and cancer. The results provide new insights into the epigenetic mechanisms of HIV that may underlie the increased risk for cancer in PWH.

Because *NLRC5* is a transcription factor regulating major histocompatibility complex (MHC) and methylation of *NLRC5* has been consistently reported to have an association with HIV infection [14,15,18,49], several studies have examined differential methylation patterns on the chromosome 6 MHC region for HIV infection [41]. Shiau *et al*. identified 28 differentially methylated regions on chromosome 6 for children on antiretroviral therapy (ART) [41]. Gross *et al*. reported significant hypomethylation of Chromosome 6 around HLA region for HIV+ compared to HIV- samples in CD4+ T cells and PBMCs [50]. Our results are consistent with these findings of hypomethylation of HLA genes with HIV infection in PBMCs and CD4 + T cells. For example, meta-EWAS in CD4+ T cells showed that five CpG sites on HLA-F (cg23892836, cg12588917, cg09296453, cg24351901, cg15331332), two CpG sites on HLA-E (cg14461571 and cg03725115), and one CpG (cg11187245) were all hypomethylated in HIV + versus HIV- across three cohorts. Together, these studies provide insights on the alteration of DNA methylation in MHC influencing immune regulation in HIV-1 pathogenesis.

Methylation changes in other immune genes and pathways beyond MHC are also linked with HIV status, ART treatment, and clinical outcomes. DMPs for HIV+ pre-ART versus HIV- were enriched in the pathways involved in immune function, such as regulation of immune system processes, immune response, and response to cytokines [14]. In our 3 cohort meta-EWAS, those pathways were also significantly associated with HIV infection (GOBP_POSITIVE_REGULATION_OF_IMMUNE_SYSTEM_PROCESS: whole blood/PBMC q value = 0.000270779; CD4+ T cell: q = 0.00288; GOBP_POSITIVE_REGULATION_OF_IMMUNE_RESPONSE, PBMC/whole blood: q value = 0.00027; CD4+ T: q = 0.01; GOBP_POSITIVE_REGULATION_OF_CYTOKINE_PRODUCTION, PBMC/whole blood: q value = 0.000282487).

More importantly, prior research has demonstrated HIV-1-induced alterations in landscape DNA methylation within monocytes and CD4 T cells during the early stages of HIV infection. A previous study by Corley *et al*. identified a large number of DMPs in monocytes (N = 22,697) but only 294 DMPs in CD4+ T-cells during acute HIV infection [13]. Interestingly, in chronic HIV infection, we observed more DMPs in CD4+ T cells compared to monocytes, suggesting dramatic changes in the epigenomic landscape between the acute and chronic stages in monocytes and CD4+ T-cells. Furthermore, Corley *et al*. observed a higher count of hypomethylated DMPs compared to hypermethylated DMPs in both monocytes and CD4+ T cells during acute HIV infection [13]. These DMPs were associated with enriched pathways involved in antiviral defense mechanisms and cell division. Conversely, our findings revealed a slightly greater number of hypermethylated DMPs than hypomethylated ones. Genes related to these DMPs were enriched in pathways associated with inflammatory functions. CD4+ T cells serve as a major host for HIV-1 latent reservoirs. Notably, DNA methylation in both host and HIV-1 epigenomes plays a critical role in establishing and maintaining HIV-1 reservoirs. It is plausible that alterations in DNA methylation within chronically infected CD4+ T cells may contribute to HIV-1 survival and latency. Together, these results show distinct methylation profiles between monocytes and CD4+ T cells in the acute versus chronic stages of HIV-1 infection.

In SIV-infected macaques and African green monkeys, only 0.5% of DMPs overlapped in CD4+ T-cells between acute and chronic HIV infection stages. This evidence suggests that distinct profiles of DNA methylation modification may occur in the different stages of HIV infection and in different cell types. Despite different pathological processes between acute and chronic HIV infection, a few DMPs near genes were the same between those reported by Corley *et al*. and the current study in monocytes (i.e., *IRF7*, *PRDM16*) and in CD4+ T-cells (i.e., *KLF7*), suggesting a small proportion HIV-associated CpG sites may persist from the acute into the chronic stage of infection.

Several previously reported genes involved in chronic HIV infection from CD4+ T-cells and from PBMC samples were replicated in this study. Of note, we observed differentially methylated CpG sites harbored in the genes involved in the Th1 signaling process (i.e., *RUNX3*, *STAT4*) in CD4+ T-cells. Several *TNF* CpG sites were reported to be hypermethylated in samples from PWH [51]. These CpG sites were also hypermethylated in the present study. A noteworthy hypomethylated DMP identified in the present study is *PEX14* cg25310676. PEX14 is involved in the control of oxidative stress and is targeted by HIV Env-mediated autophagy [52]. Expression of *PEX14* was decreased in HIV-infected CD4+ T-cells and contributed to CD4+ T-cell apoptosis [7].

The fact that many HIV-associated DMPs and genes are also involved in cancer among the four cell types is intriguing. One possibility is that HIV-1 directly induces maladaptive changes in epigenetic regulation of oncogenes. For example, several BCL family genes were significantly associated with HIV infection, *BCL9* in CD4+ T-cells, and both *BCL11B and BCL2L2* in CD4+ T-cells, B cells, NK cells, and monocytes. The BCL family plays a crucial role in the development, proliferation, differentiation, and subsequent survival of T cells and is associated with multiple cancers. *BCL9* functions in cell-cell communication in colorectal cancer [53]. *BCL11B* encodes for a protein that is a transcriptional repressor and is regulated by the NURD nucleosome remodeling and histone deacetylase complex. *BCL11B* is a hallmark of B-cell Chronic Lymphocytic Leukemia (CLL). *BLC2L2* acts as an apoptotic regulator and is linked to multiple cancers including liver cancer, lung cancer, and breast cancer [54]. On the other hand, evidence shows that chronic inflammation is involved in pathogen-induced DNA methylation changes resulting in an "epigenetic field defect" for oncogenesis. For example, our results show that several proinflammatory genes (i.e., *TNF*, *IGFBPL1*) were differentially methylated in PWH compared to PWoH. Increased inflammation is a hallmark of chronic HIV infection. Whether chronic HIV-1 results in DNA methylation of inflammatory genes contributing to cancer warrants further study. The overrepresentation of HIV-1 integration in cancer genes has been reported previously [55]. In PWH on suppressive ART, a large proportion of persisting proviruses are found in proliferating cells. One possible mechanism is to promote the proliferation and survival of latently HIV-infected cells, which in turn benefits HIV-1 persistence and reservoir expansion, undermining the host's attempts to eradicate the virus. Such interactions between HIV-1 and the host epigenome may point to underlying mechanisms of cancer development in PWH.

In this study, we applied TCA to deconvolute DNA methylation into individual cell types. Recently, several cell type deconvolution algorithms have been developed for identifying disease-associated CpG sites at the cell type resolution [27,34,56–58]. The algorithms include reference-based methods, such as CIBERSORT (CBS) [57] and EpiDISH [58] and reference-free methods such as TCA [34]. Two methods have been tested for multiple EWAS: CellDMC and TCA [34,59,60]. CellDMC incorporates an interaction term of phenotype and estimated cell-type proportion in the linear model and enables variation in effect-size to be resolved as a function of cell-type abundance in a cell-type specific manner. On the other hand, TCA uses a tensor composition analysis to learn cell-type specific DNA methylation from a typical two-dimensional bulk data (samples by methylation sites). Conceptually, TCA considers that individuals not only differ in cell type proportion, but also in methylation within a cell type. We chose to apply TCA method for this study because TCA appears to have greater power than CellDMC [34] and because TCA deconvolutes methylation values at each CpG, which is helpful for benchmarking using real methylation data in sorted cell types and for biological interpretations of the results.

We acknowledge several limitations to this study. Although the PWH in the cohort 3 are untreated (i.e., studied prior to initiation of ART), the samples from two out of three cohorts

are from PWH on long-term ART. A proportion of DMPs associated with HIV infection may be confounded by ART-associated effects. Cell type proportion was estimated based on DNA methylation, not cell count, which could impact the accuracy of deconvolution of DNA methylation for individual cell types. Also, while the results suggest the effect would be modest, TCA-deconvoluted DNA methylation profiles in each cell type may differ between the two cohorts due to differences in biospecimen collection. For cohorts 1 and 3, cell-type DNAm was deconvoluted from whole blood that included granulocytes while cell-type DNAm from the cohort 2 was deconvoluted from PBMCs, which excludes granulocytes. Computationally identified significant DMPs warrant confirmation in sorted cell types. Finally, only a small proportion of CpG sites in the methylome were investigated in this study (i.e., 450K and EPIC commercial arrays). Future studies to expand the number of CpG sites using a sequencing platform to comprehensively profile the methylome for chronic HIV infection are warranted.

In summary, leveraging a computational deconvolution approach, we identified cell-type level DMPs associated with HIV infection. The findings were enriched for genes involved in HIV pathogenesis, underscoring the important mechanisms of HIV persistence.

## Supporting information

**S1 Table. Demographic and clinical characteristics of the study cohorts.**
(XLSX)

**S2 Table. Cohort 1: Significant differentially methylated CpG positions between PWH and PWoH in whole blood in the Veteran Aging Cohort Study.**
(XLSX)

**S3 Table. Cohort 1: Top 500 significant differentially methylated CpG positions between PWH and PWoH in CD4+ T-cells from the Veteran Aging Cohort Study.**
(XLSX)

**S4 Table. Cohort 1: Significant differentially methylated CpG positions between PWH and PWoH in CD8+ T-cells from the Veteran Aging Cohort Study.**
(XLSX)

**S5 Table. Cohort 1: Significant differentially methylated CpG positions between PWH and PWoH in B cells from the Veteran Aging Cohort Study.**
(XLSX)

**S6 Table. Cohort 1: Significant differentially methylated CpG positions between PWH and PWoH in NK cells from the Veteran Aging Cohort Study.**
(XLSX)

**S7 Table. Cohort 1: Significant differentially methylated CpG positions between PWH and PWoH in monocytes from the Veteran Aging Cohort Study.**
(XLSX)

**S8 Table. Cohort 1: Top 500 significant differentially methylated CpG positions between PWH and PWoH in granulocytes from the Veteran Aging Cohort Study.**
(XLSX)

**S9 Table. Cohort 2: Significant differentially methylated CpG positions between PWH and PWoH in PBMCs from the Women's Interagency HIV Study.**
(XLSX)

**S10 Table. Cohort 2: Significant differentially methylated CpG positions between HIV-positive and HIV-negative PBMCs using all 870K CpG sites from the EPIC array from the Women's Interagency HIV Study.**
(XLSX)

**S11 Table. Cohort 2: Significant differentially methylated CpG positions between PWH and PWoH in CD4+ T-cells from the Women's Interagency HIV Study.**
(XLSX)

**S12 Table. Cohort 2: Significant differentially methylated CpG positions between PWH and PWoH in CD8+ T-cells from the Women's Interagency HIV Study.**
(XLSX)

**S13 Table. Cohort 2: Significant differentially methylated CpG positions between PWH and PWoH in B cells from the Women's Interagency HIV Study.**
(XLSX)

**S14 Table. Cohort 2: Significant differentially methylated CpG positions between PWH and PWoH in NK cells from the Women's Interagency HIV Study.**
(XLSX)

**S15 Table. Cohort 2: Significant differentially methylated CpG positions between PWH and PWoH in monocytes from the Women's Interagency HIV Study.**
(XLSX)

**S16 Table. Cohort 3: Significant differentially methylated CpG positions between PWH and PWoH in whole blood in GSE217633.**
(XLSX)

**S17 Table. Cohort 3: Top 500 significant differentially methylated CpG positions between pre-ART and post-ART PWH in whole blood in GSE217633.**
(XLSX)

**S18 Table. Cohort 3: Top 500 significant differentially methylated CpG positions between PWH and PWoH in CD4+ T-cells in GSE217633.**
(XLSX)

**S19 Table. Cohort 3: Significant differentially methylated CpG positions between PWH and PWoH in CD8+ T-cells in GSE217633.**
(XLSX)

**S20 Table. Cohort 3: Significant differentially methylated CpG positions between PWH and PWoH in B cells in GSE217633.**
(XLSX)

**S21 Table. Cohort 3: Significant differentially methylated CpG positions between PWH and PWoH in NK cells in GSE217633.**
(XLSX)

**S22 Table. Cohort 3: Significant differentially methylated CpG positions between PWH and PWoH in monocytes in GSE217633.**
(XLSX)

**S23 Table. Cohort 3: Significant differentially methylated CpG positions between PWH and PWoH in granulocytes in GSE217633.**
(XLSX)

**S24 Table. Epigenome-wide meta-analysis of PBMC DNA methylation in three combined cohorts for HIV infection, top 500.**
(XLSX)

**S25 Table. Epigenome-wide meta-analysis of CD4+ T-cell DNA methylation in three combined cohorts for HIV infection, top 500.**
(XLSX)

**S26 Table. Epigenome-wide meta-analysis of CD8+ T-cell DNA methylation in three combined cohorts for HIV infection.**
(XLSX)

**S27 Table. Epigenome-wide meta-analysis of B cell DNA methylation in three combined cohorts for HIV infection.**
(XLSX)

**S28 Table. Epigenome-wide meta-analysis of NK cell DNA methylation three combined cohorts for HIV infection.**
(XLSX)

**S29 Table. Epigenome-wide meta-analysis of monocyte DNA methylation in three combined cohorts for HIV infection.**
(XLSX)

**S30 Table. Hallmark pathways.**
(XLSX)

**S1 Text. Supplemental materials.**
(DOCX)

**S1 Fig. Cell proportion estimation prior to TCA in both VACS and WIHS cohorts.** (a) Cohort 1: VACS cell proportion. (b) Cohort 2: WIHS cell proportion. Cohort 3: GSE217633 cell proportion. Granulocytes were removed in the sequential cell-type based EWAS analyses. EWAS: Epigenome-wide Association Study; VACS: Veteran Aging Cohort Study; WIHS: Women's Interagency HIV Study.
(PDF)

**S2 Fig. Correlation analysis of top 30 Principle Components (PCs) on DNA methylation and cell type proportion in bulk cells and TCA-deconvoluted cell types from cohorts 1 and 2, for which methylation intensity data of each probe was available.** (a) VACS: -log10(p) plots for whole blood methylation and for six individual cell type methylation. (b) WIHS: -log10(p) plots for PBMC methylation and for six individual cell type methylation. PBMC: peripheral blood mononuclear cell; TCA: Tensor Composition Analysis; VACS: Veteran Aging Cohort Study; WIHS: Women's Interagency HIV Study.
(PDF)

**S3 Fig. Benchmarking TCA-deconvoluted cell-type specific DNA methylation.** Comparison of methylation β-values for the top 10,000 most variable CpG sites between the deconvoluted and the directly measured methylation. β-values for each cell type were compared between three cell types [CD4+ T-cells, CD8+ T-cells, and monocytes (CD16+)]. MC: methylation

capture sequencing; TCA: Tensor Composition Analysis.
(PDF)

**S4 Fig. Summary of EWAS prior to and following computational deconvolution in each cohort.** (a) Cohort 1: VACS, EWAS on HIV infection in whole blood. (b) Cohort 1: VACS, EWAS on HIV infection in 5 individual cell types. (c) Cohort 2: WIHS, EWAS on HIV infection in PBMCs. (d) Cohort 2: WIHS, EWAS on HIV infection in 5 individual cell types. (e) Cohort 3: GSE217633, EWAS on HIV infection in PBMCs. (f) Cohort 3: GSE217633, EWAS on HIV infection in 5 individual cell types. EWAS: Epigenome-wide Association Study; VACS: Veteran Aging Cohort Study; WIHS: Women's Interagency HIV Study.
(PDF)

**S5 Fig. EWAS on HIV in cohort 2 PBMCs using all ~870K CpG sites in the EPIC array.** EWAS: Epigenome-wide Association Study; PBMC: peripheral blood mononuclear cell.
(PDF)

**S6 Fig. Pre-ART versus post-ART EWAS in cohort 3 PBMCs.** ART: antiretroviral therapy; EWAS: Epigenome-wide Association Study; PBMC: peripheral blood mononuclear cell.
(PDF)

**S7 Fig. Characterization of epigenome-wide significant DMP from cell-type level meta-epigenomewide association analysis.** DMP: Differential Methylation Position.
(PDF)

**S8 Fig. Beta values in HIV+ and HIV- groups among three cohorts for the top meta-significant CpG sites in which beta value differed at least 5% between HIV+ and HIV- in at least two of three cohorts.**
(PDF)

**S9 Fig. GO (Gene Ontology), KEGG (Kyoto Encyclopedia of Genes and Genomes), and Wiki pathway analysis for each cell type.**
(PDF)

## Acknowledgments

The authors appreciate the support of the Veteran Aging Study Cohort Biomarker Core, Women's Interagency HIV Study, and Yale Center of Genomic Analysis. The views and opinions expressed in this manuscript are those of the authors and do not necessarily represent those of the Department of Veterans Affairs or the United States government. This work uses data provided by patients and collected by the VA as part of their care and support. We gratefully acknowledge the scientific contributions of Dr. Kendall Bryant, our Scientific Collaborator. The MWCCS is funded primarily by the National Heart, Lung, and Blood Institute (NHLBI), with additional co-funding from the Eunice Kennedy Shriver National Institute Of Child Health & Human Development (NICHD), National Institute On Aging (NIA), National Institute Of Dental & Craniofacial Research (NIDCR), National Institute Of Allergy And Infectious Diseases (NIAID), National Institute Of Neurological Disorders And Stroke (NINDS), National Institute Of Mental Health (NIMH), National Institute On Drug Abuse (NIDA), National Institute Of Nursing Research (NINR), National Cancer Institute (NCI), National Institute on Alcohol Abuse and Alcoholism (NIAAA), National Institute on Deafness and Other Communication Disorders (NIDCD), National Institute of Diabetes and Digestive and Kidney Diseases (NIDDK), National Institute on Minority Health and Health Disparities (NIMHD), and in coordination and alignment with the research priorities of the National

Institutes of Health, Office of AIDS Research (OAR). The MWCCS Principal Investigators and their funding are: Atlanta CRS (Ighovwerha Ofotokun, Anandi Sheth, and Gina Wingood), U01-HL146241; Bronx CRS (Kathryn Anastos and Anjali Sharma), U01-HL146204; Brooklyn CRS (Deborah Gustafson and Tracey Wilson), U01-HL146202; Data Analysis and Coordination Center (Gypsyamber D'Souza, Stephen Gange and Elizabeth Golub), U01-HL146193; Chicago-Cook County CRS (Mardge Cohen and Audrey French), U01-HL146245; Northern California CRS (Bradley Aouizerat, Jennifer Price, and Phyllis Tien), U01-HL146242; Metropolitan Washington CRS (Seble Kassaye and Daniel Merenstein), U01-HL146205; Miami CRS (Maria Alcaide, Margaret Fischl, and Deborah Jones), U01-HL146203; UAB-MS CRS (Mirjam-Colette Kempf, Jodie Dionne-Odom, and Deborah Konkle-Parker), U01-HL146192; UNC CRS (Adaora Adimora), U01-HL146194.

## Author Contributions

**Conceptualization:** Ying Hu, Vincent C. Marconi, Zuoheng Wang, Amy C. Justice, Bradley E. Aouizerat, Ke Xu.

**Data curation:** Mardge H. Cohen, Amy C. Justice, Bradley E. Aouizerat, Ke Xu.

**Formal analysis:** Xinyu Zhang, Ying Hu.

**Funding acquisition:** Bradley E. Aouizerat, Ke Xu.

**Investigation:** Xinyu Zhang, Ral E. Vandenhoudt, Amy C. Justice, Bradley E. Aouizerat, Ke Xu.

**Methodology:** Xinyu Zhang, Ying Hu, Zuoheng Wang, Ke Xu.

**Project administration:** Ke Xu.

**Resources:** Vincent C. Marconi, Mardge H. Cohen, Amy C. Justice, Bradley E. Aouizerat.

**Supervision:** Ying Hu, Chunhua Yan, Vincent C. Marconi, Zuoheng Wang, Amy C. Justice, Bradley E. Aouizerat, Ke Xu.

**Validation:** Ke Xu.

**Visualization:** Ral E. Vandenhoudt.

**Writing – original draft:** Xinyu Zhang, Bradley E. Aouizerat, Ke Xu.

**Writing – review & editing:** Xinyu Zhang, Ying Hu, Ral E. Vandenhoudt, Chunhua Yan, Vincent C. Marconi, Mardge H. Cohen, Zuoheng Wang, Amy C. Justice, Bradley E. Aouizerat, Ke Xu.

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
