## [Decision Letter · Decision Letter 0]

6 Dec 2023

Dear Dr. Xu,

Thank you very much for submitting your manuscript "Cell-Type Specific Epigenome-Wide DNA Methylation Analysis Unveils Distinct Methylation Patterns among Immune Cells for HIV Infection in Three Cohorts" for consideration at PLOS Pathogens. As with all papers reviewed by the journal, your manuscript was reviewed by members of the editorial board and by several independent reviewers. In light of the reviews (below this email), we would like to invite the resubmission of a significantly-revised version that takes into account the reviewers' comments.

We cannot make any decision about publication until we have seen the revised manuscript and your response to the reviewers' comments. Your revised manuscript is also likely to be sent to reviewers for further evaluation.

Sincerely,

Daniel C. Douek

Academic Editor

PLOS Pathogens

Richard Koup

Section Editor

PLOS Pathogens

Kasturi Haldar

Editor-in-Chief

PLOS Pathogens

orcid.org/0000-0001-5065-158X

Michael Malim

Editor-in-Chief

PLOS Pathogens

orcid.org/0000-0002-7699-2064

Reviewer's Responses to Questions

**Part I - Summary**

Reviewer #1: The manuscript by Zhang et al. focuses on cell-type specific epigenome-wide DNA methylation analysis among immune cells for HIV infection in three cohorts. The paper aims to identify differentially methylated CpG sites for HIV infection in immune cell types including CD4 T cells, CD8 T cells, B cells, NK cells, monocytes, and granulocytes which is a strength. The methods applies a computational deconvolution method to perform a cell-type based EWAS which is a weakness since the actual cell-type specific signal is not being directly assessed as stated in the main aim of the paper. The sample size total is 1,382 samples which is a strength and uses a widely utilized array-based method to assay DNA methylation states. The approach also analyzed the data in each cohort separately and then combined in a meta-EWAS. Overall, the study is significant for the field as it highlights the potential cell-type specific differences in epigenetic signals that are relevant for HIV infection. The major weakness is that the title and presentation of the approach should clarify that the cell-type specificity was bioinformatically inferred. I would advise to change to title to "Inferred cell-type specific...." and modify the text to reflect the inferred nature of the analyses for cell type-specificity. There also should be a discussion of how the deconvolution algorithm utilized may not reflect immune cell composition changes observed in people living with HIV.

**Part II – Major Issues: Key Experiments Required for Acceptance**

Reviewer #1: The results section states that many cell-type specific significant CpG sites were associated with HIV infection in each cohort. Line 35/36 is vague and doesn’t state how many sites. How many sites overlapped or were concordant among the three cohorts? The statement in Line 49 of the conclusion about pathogen-induced epigenetic oncogencity is weakly supported by the way the manuscript is structured and the results. Just because enrichment in that category was found doesn’t specify that over the other enriched categories. In the introduction Line 101 and 102 is not completely correct as prior studies have examined cell sorted immune cell subsets and examined differences in HIV. This is correct in lines 108 and 109. Please provide references. The Veterans Aging Cohort Study, Women’s Interagency HIV study, and public dataset GSE217633 were used as datasets. Were the Cohort 1 VACS DNA methylation data previously published or deposited in GEO? Please clarify in the Methods. A weakness is the use of data from cohort 1 that was from an older 450k array. This limits the analysis by removed over 400,000 CpGs sites that are unique to the EPIC arrays. These sites are likely more informative as they fall in gene regulatory regions. Results: The section about benchmarking deconvoluted DNA methylome data using capture bisulfite sequencing in CD4, CD8, and monocytes should be moved to supplemental or included in the methods section. The figure 2 doesn’t add anything significant to the results. Cohort 1’s analyses should clarify in text that this used the 450K data only. Please show plots of the top CpG sites based on the beta values in supplemental.For Cohort 2, what would the data look like if you used all the data for the methylation EPIC arrays? Does the frequency of CpGs increase or is there a similar number? Are the genes and pathways different? How much does the interpretation change since only the matching 450K probes are used? IN the replication cohort lines 368-370, what was the percentage of loci that were replicated in whole blood? The dataset should also be utilized to replicate the post-ART time points. This would be a more relevant validation and comparison that should be included from that dataset. Line 413 please list the examples. Lines 420-423 please provide statistics for the region enrichment analyses and show a plot or supplemental figure. The results section on line 424 should link to the cohort metadata. DO any of the participants have cancer? The section on HIV-associated DMPs analyses should specify the database used and potentially consider using multiple databases to validate the results. Sometimes these analyses are database biased since the majority have focused on cancer. The discussion should be lengthened to better review prior studies and to provide a more thorough discussion of the cell-type specific implications of these findings.

Overall the following three major issues should be addressed:

1. Presentation of beta values for the DMP examples to show biological relevance of the identified loci.

2. Reanalysis and comparison using the removed 400,000 sites from the EPIC array for COhort 2 and inclusion of Cohort 3's post-ART data for validation.

3. A comparison and discussion of how the cell-type deconvoltion algorithms and datasets changes the results.

**Part III – Minor Issues: Editorial and Data Presentation Modifications**

Reviewer #1: Figure 3 C is missing label for granulocytes. Please clarify in Lines 342 that the dataset was filtered to remove 400,000 sites to match the VACS cohort. Line 348 please provide the p values.

PLOS authors have the option to publish the peer review history of their article (what does this mean?). If published, this will include your full peer review and any attached files.

Reviewer #1: No
---

## [Decision Letter · Decision Letter 1]

20 Feb 2024

Dear Dr. Xu,

We are pleased to inform you that your manuscript 'Computationally Inferred Cell-Type Specific Epigenome-Wide DNA Methylation Analysis Unveils Distinct Methylation Patterns among Immune Cells for HIV Infection in Three Cohorts' has been provisionally accepted for publication in PLOS Pathogens.

Best regards,

Daniel C. Douek

Academic Editor

PLOS Pathogens

Richard Koup

Section Editor

PLOS Pathogens

Michael Malim

Editor-in-Chief

PLOS Pathogens

orcid.org/0000-0002-7699-2064

Reviewer Comments (if any, and for reference):

Reviewer's Responses to Questions

**Part I - Summary**

Reviewer #1: Authors addressed major concerns and manuscript has improved to reflect a major contribution for the field in regards to distinct DNA methylation patterns among immune cells for HIV infection.

**Part II – Major Issues: Key Experiments Required for Acceptance**

Reviewer #1: None

**Part III – Minor Issues: Editorial and Data Presentation Modifications**

Reviewer #1: None

PLOS authors have the option to publish the peer review history of their article (what does this mean?). If published, this will include your full peer review and any attached files.

Reviewer #1: No

---

## [Editor Report · Acceptance letter]

5 Mar 2024

Dear Dr. Xu,

We are delighted to inform you that your manuscript, " Computationally Inferred Cell-Type Specific Epigenome-Wide DNA Methylation Analysis Unveils Distinct Methylation Patterns among Immune Cells for HIV Infection in Three Cohorts ," has been formally accepted for publication in PLOS Pathogens.

Best regards,

Michael Malim

Editor-in-Chief

PLOS Pathogens

orcid.org/0000-0002-7699-2064